# An Emerging Role of Defective Copper Metabolism in Heart Disease

**DOI:** 10.3390/nu14030700

**Published:** 2022-02-07

**Authors:** Yun Liu, Ji Miao

**Affiliations:** 1Key Laboratory of Molecular Target & Clinical Pharmacology and the State & NMPA Key Laboratory of Respiratory Disease, School of Pharmaceutical Sciences & The Fifth Affiliated Hospital, Guangzhou Medical University, Guangzhou 511436, China; liuyun195@gzhmu.edu.cn; 2Division of Endocrinology, Boston Children’s Hospital, Boston, MA 02115, USA; 3Department of Medicine, Harvard Medical School, Boston, MA 02115, USA

**Keywords:** copper chaperone, copper transporter, copper deficiency, heart disease

## Abstract

Copper is an essential trace metal element that significantly affects human physiology and pathology by regulating various important biological processes, including mitochondrial oxidative phosphorylation, iron mobilization, connective tissue crosslinking, antioxidant defense, melanin synthesis, blood clotting, and neuron peptide maturation. Increasing lines of evidence obtained from studies of cell culture, animals, and human genetics have demonstrated that dysregulation of copper metabolism causes heart disease, which is the leading cause of mortality in the US. Defects of copper homeostasis caused by perturbed regulation of copper chaperones or copper transporters or by copper deficiency resulted in various types of heart disease, including cardiac hypertrophy, heart failure, ischemic heart disease, and diabetes mellitus cardiomyopathy. This review aims to provide a timely summary of the effects of defective copper homeostasis on heart disease and discuss potential underlying molecular mechanisms.

## 1. Introduction

Despite decades of efforts to understand its pathogenesis, heart disease remains a leading cause of mortality and places a growing burden on healthcare systems worldwide. Copper is an essential trace metal micronutrient that has been previously overlooked, but has recently gained attention and become an emerging player in the development of heart disease. Although it is less abundant than other metals such as iron and zinc, copper is widely utilized as a catalytic or structural cofactor by enzymes and proteins that are highly relevant to cardiac physiology and pathology. These copper-binding proteins include cytochrome c oxidase (CCO), superoxide dismutase (SOD), metallothionein (MT), ceruloplasmin (CP), and lysyl oxidase (LOX), which regulate mitochondrial respiration, antioxidant defense, iron metabolism, and connective tissue crosslinking [1]. Increasing evidence has demonstrated that dysregulation of copper homeostasis causes heart disease.

Copper exists in two ionic forms in the body, namely, cuprous Cu^+^, which is dominant in the intracellular reductive environment, and cupric Cu^2+^, which is dominant in the extracellular oxidative environment. In mammals, copper is exclusively absorbed from diets and water by enterocytes in the small intestine via the Cu^+^-specific copper transporter 1 (CTR1). The reduction of Cu^2+^ to Cu^+^ is necessary for its entry into cells and is mediated by copper reductases, including members of the plasma membrane-bound six transmembrane epithelial antigen of the prostate (STEAP) family [2]. The latest study by Kurdistani et al. identified a novel copper reductase enzyme, the histone H3-H4 tetramer, which binds to Cu^2+^ and catalyzes its reduction to Cu^+^ in *Saccharomyces cerevisiae* yeast to maintain the function of the electron transport chain in mitochondria [3]. After entering interstitial fluid, copper initially binds to albumin or transcuprein, travels through the portal circulation, and is taken up by the liver via hepatic CTR1. Copper in the liver is then incorporated into CP, a major plasma copper-binding protein responsible for carrying >90% of copper in the circulation [4]. Copper-loaded CP delivers copper to extra-hepatic tissues, and excess copper returns to the liver for excretion into bile through hepatic copper-transporting ATPase 2 (ATP7B). Intracellular copper transportation and involved copper-binding proteins and chaperons are depicted in Figure 1.

Copper is vital for cellular functions, and excess copper is toxic. Therefore, the distribution and amount of bioavailable copper must be tightly controlled to meet metabolic requirements, while minimizing potential toxicity of excess copper. Impaired functions of copper transporters, defects in copper-dependent enzymes, and chronic copper deficiency cause heart diseases [5,6,7,8,9,10]. While the recommended daily allowance (RDA) of copper is 0.9 mg/day, the recommended optimal intake is 2.6 mg/day [11]. The copper requirement varies between individuals and depends on age, pregnancy, sex, health status, and other factors. For example, RDA is 340 µg/day for 1–3-year-old children and increases to a minimum of 1 mg/day during pregnancy [12]. Excessive daily zinc intake competes with copper for absorption by enterocytes in the small intestine and, therefore, decreases copper intake [13,14]. Diseased states, including hypertension, ischemic heart disease (IHD), heart failure (HF), nephrotic syndrome, and celiac disease, often cause copper deficiency, which increases demands for daily copper intake [15,16,17,18,19,20,21,22].

Although a trace amount of copper is required daily, copper deficiency is common because the amount of copper in modern diets has decreased during the last several decades. Western-style diets enriched in saturated fat and simple sugars, particularly fructose, inhibit small intestinal copper absorption and, thus, contribute little to the daily dietary requirement for copper [11,23]. In addition, changes in farming methods have decreased the copper content of soil and, thus, in produce [24]. In fact, the National Health and Nutrition Examination Survey (NHANES III, 2003) revealed that more than 80% of 103,655 people studied in the US received a lower amount of copper than the RDA from their diet [25]. Similarly, dietary copper intake was lower in the National Diet and Nutrition Survey (NDNS) from 2000/01 [26] than in that from 1986/87 [27] in the UK.

In the past decades, studies from dozens of laboratories across the globe have revealed that copper plays a major role in maintaining normal heart morphology and function. A recent review summarizes the association between copper deficiency and IHD in detail [15]. Another summarizes the link between copper transporters and chaperones and cardiovascular disease [22]. In this review, we thoroughly summarize current knowledge about the roles of copper-dependent enzymes, copper transporters/chaperones, and copper deficiency in heart diseases, including cardiac hypertrophy, HF, IHD, and diabetes mellitus (DM) cardiomyopathy, focusing on detailing the molecular mechanisms and providing both preclinical and clinical evidence.

## 2. Copper Chaperones and Heart Physiological and Pathological Processes

### 2.1. CCO

Copper-dependent CCO is the mitochondrial respiratory chain complex IV, which contains copper and heme as required co-factors and plays a critical role in oxidative phosphorylation. Of the 13 subunits of mammalian CCO, three mitochondria-encoded DNA subunits (I, II, and III) contain copper and heme in their active sites and constitute the catalytic core of the oxidase complex. Several copper chaperones deliver copper to CCO, including CCO copper chaperone 11 (COX11), COX17, COX19, and COX23 and synthesis of CCO 1 (SCO1) and SCO2. Thus, copper deficiency reduces CCO activity and the mitochondrial respiratory capacity in the heart, for which its functions heavily rely on intact mitochondrial respiration [8,28,29,30,31,32,33,34]. Copper deficiency causes cardiac hypertrophy by impairing mitochondrial function and energy production, evidenced by increases in mitochondrial compensatory biogenesis and size and mitochondrial ultrastructural deteriorations, as well as a decrease in the number or loss of cristae [19,35].

Although the causative role of copper deficiency in decreasing CCO activity was documented as early as 1939 by Schttltze [36], the key players have only recently been identified based on human genetic studies and loss-of-function animal studies. Patients with the common E140K mutation in SCO2 were reported to die of HF due to fetal/infantile hypertrophic cardiomyopathy (HCM). This mutation is adjacent to the proposed SCO2 copper-binding motif C133xxxC137 (x denotes any amino acid) [10,37,38,39,40]. It converts negatively charged glutamate to positively charged lysine and, thus, displaces copper from the copper-binding domain and subsequently reduces CCO activity. Although patients with the homozygous E140K mutation exhibit a milder phenotype with later onset HCM and relatively slow progression of HF, patients with the heterozygous E140K mutation and other SCO2 missense point mutations rapidly develop HCM and HF after birth. For example, two patients carrying Q53X and E140K mutations in SCO2 were diagnosed with HCM at 6 and 10 weeks of age, respectively, and died at 11 and 24 weeks of age, respectively [38]. By contrast, a patient with the homozygous E140K mutation in SCO2 was diagnosed with HCM at 13 months of age and rapidly progressed to moderate HCM at 21 months of age. Subcutaneous injections of copper-histidine to this patient starting when she was 23 months old for 2 months failed to improve HCM, and interventricular septal thickness increased to 1.5 cm by the end of the treatment [10]. However, compellingly, oral copper administration at a dose of 140 μg/day starting when the patient was 25 months old remarkably ameliorated HCM, and interventricular septal thickness was reduced to 0.9 cm when she was 39 months old. Although this patient died from severe pneumonia at 42 months of age, she showed significant improvements of HCM and survived longer than all previously reported patients with the same mutation who died before 25 months of age [10].

The G132S mutation in the juxtamembrane region of SCO1 (a SCO2 paralog with complementary functions) is also associated with early onset HCM. This mutation reduces protein stability of SCO1 by preventing its oligomerization and, thus, decreases CCO activity in skeletal muscle [41]. A patient with this mutation had detectable left ventricle hypertrophy at 2 months of age and died at 6 months of age due to HF with prominent cardiac concentric hypertrophy. Mice with cardiomyocyte-specific ablation of *Sco1* show dilated cardiomyopathy with both copper deficiency and a reduction in CCO activity by 60% [42]. Consistently, copper deficiency also reduces protein expression of mitochondrial CCO subunits (MT-COs), which contributes to reduced CCO activity and subsequent cardiomyopathy. In rat neonates with copper deficiency induced by dietary copper restriction, MT-CO1 protein expression decreased by 40% starting at postnatal day (PND) 10, and MT-CO4 protein expression decreased by 20% starting at PND 21, paralleling decreased CCO activity in cardiac mitochondria in these rats [28]. Post-weaning copper repletion in these pups for 6 weeks failed to restore MT-CO1 expression and CCO activity in cardiac mitochondria, and copper repletion for 9 months failed to restore the copper level comparable to that of control rats [28,30]. These data highlight that copper is essential for cardiac development, and that reductions in cardiac CCO activity caused by copper deficiency during the perinatal period have a profound impact on heart functions that lasts into adulthood.

In addition to copper chaperones, the roles of their regulators in heart disease have also been described. CCO assembly factor 6 (COA6) participates in CCO complex assembly [43,44]. Thus, a COA6 mutation, W66R, in humans is associated with mitochondrial complex IV deficiency in the heart and subsequently causes cardiac hypertrophy [8]. Fibroblasts from a patient with this loss-of-function mutation of COA6 showed an increase in protein turnover of mitochondrial complex IV subunits, including MT-CO1, 2, and 4. Compellingly, the addition of copper chloride to these fibroblasts for 7 days partially restored protein expression levels of complex IV and its subunits [8]. Further mechanistic studies revealed that COA6 interacts with SCO2 and is required to maintain COX2 protein stability. The absence of COA6 results in rapid MT-CO2 protein turnover and concomitant reductions in CCO levels and activity in yeast [7]. A further sequencing study revealed that a conserved Cx_9_Cx_n_Cx_10_C (x denotes any amino acid) motif, which includes W66, in COA6 is crucial for copper-dependent mitochondrial respiration [8]. In addition, patients with mutations of this conserved Cx_9_Cx_n_Cx_10_C motif in COA6 protein, such as the missense mutation W59C or the nonsense mutation E87X, all had disrupted COA6 functions and subsequently developed severe mitochondrial respiratory chain disease (MRCD) with cardiac hypertrophy and HF that resulted in premature death at 1 year of age [45]. Consistently, *coa6* knockdown in zebrafish embryos causes heart development defects that parallel those observed in humans [45], highlighting the importance of COA6 in cardiac development and pathology. Taken together, these data demonstrate the key roles of the CCO complex, its copper chaperones, and regulators of these chaperones in cardiac mitochondrial function and the development of copper deficiency-dependent cardiomyopathy. Targeting these factors may be a promising approach to treat cardiac hypertrophy and HF. The impacts of mutations/deletions of SCO1, COA6, and other copper-binding proteins in humans and preclinical animal models discussed in this article are listed in Table 1.

### 2.2. SOD

The copper- and zinc-containing SOD1 enzyme is critical for antioxidant defense by catalyzing dismutation of the deleterious superoxide radical (O_2_^−^) to molecular oxygen or hydrogen peroxide, which is in turn reduced to water by other enzymes [59]. The copper chaperone for SOD (CCS) binds to Cu^+^ via its N- or C-terminus, while it interacts with SOD1 via the central domain. Delivery of copper to SOD1 by the Cu-CCS complex permits formation of disulfide bonds in SOD1, which are required for its enzymatic activity and for the prevention of its misfolding, aggregation, and inactivation [60]. Mice with global *Ccs* deletion show markedly reduced SOD1 activity due to impaired copper incorporation into SOD1, supporting the important role of CCS in the regulation of SOD1 activity [61].

Compared with wild-type mice, *Sod1*-knockout mice show exacerbated oxidative stress. Although no cardiac function data were presented, Zhu et al. reported that due to excessive oxidative stress in the ischemic heart, *Sod1*-knockout mice are vulnerable to cardiac injury after the induction of acute myocardial ischemia by occlusion of the anterior descending branch of the left coronary artery [46]. In addition, copper deficiency, via a reduction in SOD1 activity in endothelial cells, results in a reduction in nitric oxide (NO) and an elevation of superoxide anions, which in turn impairs endothelial function and adversely affects mouse embryonic heart development, resulting in a swollen heart and pericardial effusion [62]. Although copper-dependent SOD1 is critical for antioxidant defense, there are no reports of a link between SOD1 mutations and heart disease in humans.

Manganese-containing SOD2 (MnSOD) localizes to mitochondria and serves as the first line of defense against mitochondrial respiration-generated oxidative stress. Although SOD2 does not bind to copper, it indirectly regulates the activity of copper-containing SOD1. *Sod2*-deficient mice show increased release of superoxide anion radical derivatives and impaired SOD1 activity, which causes HF [63]. These mice exhibit myocardial damage, with enlarged mitochondria, loss of cristae, and fewer myofilaments, as well as lipid peroxidation and activation of apoptosis. A recent exon sequencing study revealed that the homozygous G181V missense mutation in SOD2 causes severe cardiomyopathy in human newborns, manifesting as severe biventricular dilation and a decreased left ventricular ejection fraction. A patient bearing this mutation died at 4 days of age. This mutation disrupts the mitochondrial superoxide scavenging activity of SOD2, and subsequently results in the rapid development of HF and death [59]. These studies provided insights into the molecular mechanisms by which copper defects and superoxide induce cardiac injury.

Copper-bound SOD3, which is primarily expressed in blood vessels and is extracellularly localized, has also been linked to heart disease. Activity and/or expression of copper-bound SOD3 was reported to be decreased in animal models and humans with hypertension, HF, and coronary heart disease [64,65,66]. *Sod3* deletion strikingly increases the expression of collagen and matrix metalloproteinase-2 and -9 and production of superoxide anions in mice subjected to transaortic constriction (TAC). Accordingly, *Sod3* deletion in mice worsens cardiac hypertrophy, left ventricular dilation, and fibrosis induced by pressure overload [48] and increases myocardial apoptosis, fibrosis, and inflammation induced by doxorubicin [49]. A SOD3 R231G variant, which processes less antioxidant ability, was reported to be positively associated with IHD, myocardial infarction, and HF in diabetic subjects [50,51,52,53,54]. The rs7655372 variant of the *SOD3* gene was associated with a significantly increased risk of ischemic stroke in the Chinese Han population of Dali City [67]. By contrast, a protective T-allele of the rs2284659 variant in the promoter region of the *SOD3* gene was negatively associated with the incidence of myocardial infarction and cardiovascular and all-cause mortality in both type 1 and type 2 diabetic patients [52]. Consistently, Oster et al. showed that cardiac copper levels positively correlated with the cardiac ejection fraction in 27 patients with coronary heart disease who underwent coronary artery bypass surgery [68]. This suggests that restoration of copper homeostasis in the heart is beneficial to heart disease, likely by improving antioxidant defense and mitochondrial function.

### 2.3. MTs

MTs are cysteine-rich, low molecular weight proteins that bind to copper and serve as intracellular copper scavengers. The formation of copper-thiolate clusters in MTs sequesters excess copper in cells and thereby minimizes copper toxicity [69]. Thus, deletion of MT1/2 renders mice hypersensitive to copper toxicity [70]. Accordingly, MT1/2-knockout mice show severe cardiac dysfunction, oxidative stress, and cardiac fibrosis, which is further exacerbated during intermittent hypoxia, whereas mice overexpressing cardiac-specific MT-IIa are protected against intermittent hypoxia-induced cardiomyopathy [47]. Although genetic cardiac-specific MT-IIa overexpression in mice failed to prevent the initiation of cardiac hypertrophy induced by copper deficiency, it inhibited progression from cardiac hypertrophy to HF during copper deficiency. This is likely due to attenuation of cardiac lipid peroxidation and a reduction in natriuretic peptide A (ANP) production and, thus, in ANP-induced myocardial apoptosis [71]. These cardiac-specific MT-IIa-overexpressing mice are resistant to doxorubicin-induced cardiotoxicity and atrial contractions due to the prevention of the deterioration of mitochondrial morphology and reduction in creatine phosphokinase levels [72]. These data demonstrate the importance of MTs in cardiac function, oxidative stress, and apoptosis, as well as the protective role of cardiac MTs in the pathologic progression of cardiac hypertrophy to HF.

### 2.4. CP

CP carries more than 90% copper in plasma and, thus, is critical for maintaining activities of copper-dependent enzymes, including SOD1 and 3, and thus the removal of oxygen radicals. CP is also a ferroxidase that is important to mobilize iron. Thus, copper deficiency that reduces CP oxidative activity may indirectly impact heart disease via dysregulation of iron homeostasis [73]. In addition, CP is an oxidase for NO that converts NO to nitrite in vivo [74,75]. Circulating CP is negatively associated with NO bioavailability, presumably due to increased conversions of NO, and enhanced oxidative stress in turn adversely impacts heart function [56].

Although the detailed molecular mechanism is unclear, elevated circulating CP is associated with DM, obesity, dyslipidemia, atherosclerosis, IHD, and mortality [76,77,78,79,80,81,82,83]. Numerous clinical studies show that circulating CP positively correlates with the risk of HF and mortality and is an independent and robust predictor of cardiovascular disease, HF, and mortality [56,78,79,84]. Two independent genome-wide association studies derived from the Cleveland Clinic GeneBank Study (4177 patients) and the Atherosclerosis Risk in Communities Study (ARIC study, 9240 patients) suggested that a single locus (rs1307255) on chromosome 3 in the *CP* gene increases CP levels [55,56]. However, both studies demonstrated no association between the rs1307255 variant and the incidence of HF. Nonetheless, these studies revealed that circulating CP levels are associated with major adverse cardiovascular events, including myocardial infarction, stroke, HF, and all-cause mortality. An increase in one standard deviation of circulating CP (79 mg/L) was associated with a 14% increase in the risk of HF [56]. Greater efforts are needed to understand whether elevated circulating CP plays a causative and pathogenic role in heart disease and the associated molecular mechanisms.

### 2.5. LOX

The copper-dependent LOX enzyme is critical to catalyze lysine-derived crosslinking of collagen and elastin fibrils in the extracellular matrix. Inhibition of collagen and elastin crosslinking reduces the tensile strength and elastic properties of connective tissues, which results in a failure to maintain normal cardiac contraction and the development of concentric cardiac hypertrophy [85,86]. Therefore, reduced LOX activity in hearts of copper-deficient rats resulted in an abnormal connective tissue network and systolic dysfunction [87,88]. The collagen network of the myocardium is primarily composed of type I collagen, which is a large-diameter collagen fiber, and type III collagen, which is a relatively small-diameter collagen fiber. Collagen fibers have high tensile strength. Thus, small changes in the concentration, composition, and diameter of collagen and the degree of crosslinking profoundly affect the mechanical properties of the heart [89,90]. Tissues containing predominantly type I collagen and/or with a high degree of crosslinking are stiffer than tissues mainly containing type III collagen and/or with less crosslinking. Copper deficiency reduces the type I-to-type III collagen ratio and, thus, collagen and elastin crosslinking, presumably due to decreased LOX activity. Consistently, in hearts of copper-deficient rats, the type III-to-type I collagen ratio was significantly increased by 5-fold after feeding a copper-deficient diet for 6 weeks and remained 2–3-fold higher from 8 to 12 weeks of feeding [91].

By contrast, an increased level of LOX is associated with excess crosslinking of type I and III collagens in patients with enhanced myocardial stiffness and HF [92,93]. The elevated type III-to-type I collagen ratio results in reduced stiffness and strength of collagen fibrils. These fibrils are also resistant to proteolytic enzymes, resulting in increased collagen deposition in the extracellular space, which initiates pathologic changes in the heart, including cardiac hypertrophy and myocardial fibrosis. Although no reports have linked LOX genetic variance to cardiomyopathy, a single nucleotide polymorphism in the *LOX* coding region, G473A, which causes the non-conservative R158E mutation in LOX protein, is linked to oral submucous fibrosis [94]. However, further loss- or gain-of-function animal studies and human genetic studies are needed to validate the role of LOX in cardiac fibrosis and to determine whether LOX is a target for the treatment of cardiomyopathy.

## 3. Copper Transporters and Heart Physiology and Pathology

The molecular mechanisms by which copper deficiency promotes cardiomyopathy were also revealed by studies defining the roles of copper transporters. The cellular copper level is precisely coordinated by its uptake and efflux via various copper transporters, including CTR1, CTR2, ATP7A, and ATP7B (Figure 1). CTR1 is the major high-affinity copper importer that localizes to the plasma membrane and endosomes, whereas CTR2 is a low-affinity copper importer that localizes to endosomes and lysosomes. Ablation of CTR2 reduces the generation of truncated CTR1 lacking a copper-binding echo domain and, thus, increases tissue copper contents [95]. Compared with controls, cardiomyocyte-specific *Ctr1*-knockout mice show low cardiac copper levels and severe cardiomyopathy with cardiac hypertrophy, endocardial fibrosis, and disordered sarcomere arrays. These mice also display defects in copper homeostasis in plasma and the liver, with a decrease in hepatic copper content and an increase in serum copper concentrations due to upregulation of ATP7A expression in the liver and small intestine [5]. Consistently, intestinal epithelial cell-specific *Ctr1*-knockout mice show diminished dietary copper absorption and, thus, reduced cardiac copper uptake. These mice exhibit cardiac hypertrophy, with an increased heart-to-body mass ratio and enlarged mitochondria harboring disordered cristae and excess large vacuoles. These abnormalities can be partially rescued by postnatal copper administration [57]. A copper-deficient diet reduces heart CTR2 protein expression by 46% compared with a copper-adequate diet in rats, demonstrating a potential association of a reduction in CTR2 in cardiac copper deficiency and heart disease [96]. However, further in vivo and in vitro studies are required to determine the role of CTR2 in heart disease.

ATP7A and ATP7B are copper exporters belonging to the P-type ATPase family and contain an ATP hydrolysis domain to provide energy for copper trafficking. ATP7A is ubiquitously expressed, with the exception of the liver in normal states, whereas ATP7B is predominantly expressed in the liver and some regions of the brain, placenta, kidney, and mammary tissue [97]. Under normal conditions, ATP7A and ATP7B localize to the trans-Golgi network (TGN), where they supply copper to copper-dependent enzymes in the secretory pathway. When cytosolic copper level rises, ATP7A or ATP7B interacts with the p62 subunit of dynactin (DNCT4) and traffics to endosome-like vesicles and then to the plasma membrane, pumping excess copper into the extracellular space, or into bile in the case of the liver, to reduce the intracellular copper level [98]. By contrast, when the intracellular copper level is low, ATP7A or ATP7B recycles to TGN and transports copper from the cytoplasm into the Golgi. In Menkes disease, the loss-of-function of ATP7A impairs apical absorption of copper in enterocytes; thus, this disease is characterized by accumulation of excess copper in the small intestine and copper deficiency elsewhere. A clinical study of 95 Menkes disease patients demonstrated a 4-fold increase in the frequency of congenital heart disease (4.2%) compared with a prevalence of 1% in the general population [58]. These results confirm that dysfunction of copper transporters causes cardiomyopathy due to an imbalance of cellular copper.

## 4. Copper Deficiency and Heart Disease

### 4.1. Cardiac Hypertrophy and HF

Progression from cardiac hypertrophy to HF was divided into three stages by Meerson [99]. In the first early developing stage, the metabolic requirement of the body exceeds cardiac output, and this stage is characterized by compensatory increased protein synthesis, mitochondrial biogenesis, and enlargement, followed by increased growth of myofibrils. In the second compensatory stage, cardiac output is induced to sustain the increase in cardiac mass and performance, and this stage is characterized by an increase in myofibril growth but impaired contractility. In the last decompensated stage, the mitochondrial-to-myofibrillar ratio decreases with ventricular dilation and a decline in cardiac output.

#### 4.1.1. Cardiac Hypertrophy

The most notable early response of the heart to copper deficiency is the initiation and progression of cardiac hypertrophy [19,71,100,101]. Cardiac hypertrophy is an independent risk factor for the development of heart diseases, including acute myocardial infarction, arrhythmia, valvular heart disease, and HF. The relationship between copper deficiency, mitochondrial defects, and cardiac hypertrophy was first described in 1970 by Goodman et al. They showed that enlargement of the mitochondrial compartment is a major contributor to cardiac hypertrophy in copper-deficient rats [35]. Copper deficiency-induced cardiac hypertrophy is concentric, resembling the effects of pressure overload, and is manifested by a thickening of the ventricular wall and interventricular septum with no change or a slight decrease in the size of the ventricular lumen. Anemia was also speculated to be a contributor to cardiac hypertrophy with copper deficiency [102]. However, other studies demonstrated that hypertrophy in copper deficiency can occur before and in the absence of anemia [103,104]. In fact, the degree of anemia appears to have no association with the degree of cardiac hypertrophy [35,105,106]. Rather, decreased CCO activity and ATP synthase function, with compensatory enlargement of mitochondria and mitochondrial biogenesis, contribute to cardiac hypertrophy [19].

In addition, Kang et al. demonstrated that in a mouse model of ascending aortic constriction-induced cardiac hypertrophy, copper supplementation attenuates cardiac hypertrophy partly by restoring expression of myocardial vascular endothelial growth factor (VEGF) and angiogenesis. Mechanistically, direct binding of CCS to hypoxia inducible factor 1α (HIF1α) mediates copper-dependent increases in HIF1α transcription activity and subsequently *Vegf* gene expression in cultured cardiomyocytes [107]. A subsequent study of rat cardiac H9C2 cells by the same group supported the critical role of VEGF in copper deficiency-induced cardiac hypertrophy [108]. Treatment with 5 μM copper sulfate attenuates hydrogen peroxide-induced cell hypertrophy, which is blunted by treatment with an anti-VEGF antibody in these cells. VEGF functions as a key regulator of induction of myocardial angiogenesis by promoting endothelial cell differentiation and migration and, thus, is crucial to sustain heart function. Numerous studies confirmed that VEGF is critical for heart disease and that inhibition of VEGF results in a transition from compensated cardiac hypertrophy to decompensated HF [109,110,111]. Indeed, although studies determining the role of VEGF in vivo using genetically modified mice are lacking, low VEGF expression is associated with HF in humans [112]. Aharinejad et al. showed that cardiac expression of VEGF was reduced in patients with dilated cardiomyopathy and HF. Therefore, activation of VEGF specifically in cardiomyocytes may be a promising approach to treat copper deficiency-induced cardiac hypertrophy. Subsequently, Kang et al. showed that hypertrophic cardiomyocytes re-enter the cell cycle and undergo mitosis and proliferation, evidenced by increased Ki-67 and phosphorylated histone H3 levels, which contribute to the development of copper deficiency-induced cardiac hypertrophy [113]. The contribution of copper deficiency to cardiac hypertrophy is clear; however, additional studies are warranted to elucidate the underlying molecular mechanisms, given that copper is involved in a variety of critical cellular processes.

In addition, the association between copper deficiency and hypertension has been reported in humans [17,114] and murine models [115,116,117]. Prolonged high blood pressure is a risk factor for cardiac hypertrophy and can result in cardiac hypertrophy by increasing the cardiac workload to meet the body’s requirement. Several lines of evidence suggest that copper-regulatory proteins play a role in regulation of blood pressure. As a type of compensatory regulation, angiotensin II upregulates expression and activity of antioxidant 1 copper chaperon (ATOX1), which increases transcription and activity of extracellular SOD3 in aortas by acting as a copper-binding transcription factor and chaperone for SOD3. Therefore, *Atox1*-knockout in mice blunts the induction of SOD3 by angiotensin II and, thus, exacerbates increased vasoconstriction in mesenteric arterioles and hypertension induced by angiotensin II [118]. It was also reported that angiotensin II promotes APT7A-SOD3 interaction and delivery of copper to SOD3 in cultured vascular smooth muscle and mouse aortas. Consistently, *Atp7a*-knockout worsens angiotensin II-induced hypertension by blunting SOD3 activity in mice [119].

#### 4.1.2. HF

Copper deficiency deleteriously affects all stages of progression from cardiac hypertrophy to HF. The hallmarks of copper deficiency-induced HF are diastolic dysfunction and a blunted response to β-adrenergic stimulation. Kang et al. showed that diet-induced copper deficiency in mice for 5 weeks starting from PND 3 results in systolic and diastolic dysfunction, including a significant 17% decrease in left ventricular peak systolic pressure (LVPSP), a ~50% decrease in the maximum rate of the rise of left ventricular pressure (+dP/dt) and in the maximum rate of the decline of left ventricular pressure (−dP/dt), as well as increases in left ventricular end diastolic pressure (LVEDP) and the duration of relaxation by 115% and 23%, respectively. Furthermore, hearts of copper-deficient mice have a blunted response to isoproterenol, a β-adrenergic agonist [120]. After 9 or 15 months of copper-restricted diet feeding, rats show cardiac diastolic and systolic dysfunction, evidenced by a blunted response of +dP/dt, −dP/dt, and LVEDP to isoproterenol [121]. Feeding a copper-adequate diet for 4 weeks to diet-induced copper-deficient mice completely restores cardiac diastolic and systolic function and the response to β-adrenergic stimulation [122], suggesting that the cardiac response to β-adrenergic stimulation requires copper.

The molecular mechanisms by which copper deficiency induces HF also include perturbation of cellular calcium homeostasis and elevated NO. Intracellular calcium homeostasis is regulated by sarcoplasmic endoplasmic reticulum calcium ATPase (SERCA), sodium/calcium exchanger (NCX), and ryanodine receptors (RyRs) [123]. Kang et al. showed that dietary copper deficiency significantly changes expression of calcium cycling genes in mouse heart, including a decrease in the L-type calcium channel, which causes a release of calcium from the sarcoplasmic reticulum through RyRs and potassium-dependent NCX. Although cardiac function data were lacking, copper repletion via copper-adequate diet feeding compellingly normalized expression of these calcium regulatory genes in copper-deficient mice [124]. In addition, copper deficiency impairs cardiac contractile function and calcium homeostasis by elevating expression of phospholamban (PLB), which inhibits SERCA2a-dependent calcium uptake [125]. Calcium cycling is complex and requires multiple types of regulation, and additional calcium channels and regulatory proteins are reportedly involved in cardiac function and pathologies, including calcium release-activated calcium channel protein 1 (Orai1), stromal interaction molecule 1 (STIM1), and transient receptor potential cation channel, subfamily M, member 7 (TRPM7) [126,127,128]. It will be of great interest to determine whether copper regulates these proteins and their upstream regulators.

Copper deficiency reduces NO production by inhibiting SOD activity in endothelial cells and consequently impairs endothelial function [68]. However, the effect of copper deficiency on NO production in the heart differs from that in endothelial cells. Sarri et al. suggested that in the rat heart, copper deficiency enhances NO production by increasing the protein levels of endothelial nitric oxide synthase (eNOS) and inducible nitric oxide synthase (iNOS). Copper deficiency also upregulates cyclic guanosine monophosphate (cGMP) and, thus, contributes to impaired cardiac contraction in rats [129,130,131]. These studies highlight the importance of NO in copper deficiency-induced heart disease. The molecular mechanisms underlying the role of NO in regulation of heart pathology by copper deficiency warrant further investigation.

However, it is still debated how copper deficiency affects cardiac contractility. Saari et al. attributed the increased contractility of cardiomyocytes isolated from copper-deficient rats to the induction of a compensatory survival pathway through an enhancement of sensitivity to insulin-like growth factor-1 (IGF-1) [132,133]. Nonetheless, it is clear that increasing the amount of fat in the diet worsens the impaired cardiac contractile function induced by a marginal copper-deficient diet in rats [125,134,135], presumably due to dysfunctional calcium cycling in cardiomyocytes and a blunted response to β-adrenergic receptor activation [136,137,138].

### 4.2. IHD

Copper deficiency results in IHD, commonly called coronary heart disease [16,18,139], via dyslipidemia through multiple mechanisms involved in dysregulation of cholesterol, fatty acid, triglyceride, and lipoprotein metabolism [140,141,142]. Copper deficiency results in the accumulation of free fatty acids in the heart and liver [142,143,144]. One possible explanation is that copper deficiency increases fatty acid synthesis by increasing nuclear localization of mature sterol regulatory element-binding transcription factor 1 (SREBP1) and thereby increasing expression of de novo lipogenic genes, including fatty acid synthase (FASN), in rat livers [143]. Copper deficiency also suppresses fatty acid oxidation and utilization. Transcriptomic analysis of the small intestine of copper-deficient rats showed that copper deficiency suppresses the expression of mitochondrial and peroxisomal fatty acid β-oxidation genes, including carnitine palmitoyltransferase 1 (*Cpt1*); L-3-hydroxyacyl CoA dehydrogenase (*Hadhb*); acyl-CoA synthetase long-chain family member 1 and 3 (*Acsl1* and *Acsl3*); δ-2-enoyl-CoA isomerase (*Peci*); and carnitine-octanoyl transferase (*Crot*) [145]. Mao et al. reported that, in hearts of copper-deficient rats, the medium-chain acyl-CoA dehydrogenase (MCAD) transcript level is low, and this contributes to cardiac lipid accumulation [146]. Dietary copper supplementation (45 mg/kg) increased fatty acid uptake and oxidation by upregulating gene expression of fatty acid transport protein (*Fatp*), fatty acid-binding protein (*Fabp*), *Cpt1*, and *Cpt2* in the liver, skeletal muscle, and adipose tissue of rabbits [147]. In addition, copper deficiency alters the fatty acid composition toward a profile that is positively associated with IHD incidence, with an increase in medium- and long-chain saturated fatty acids in various tissues and in the circulation [148,149], while copper supplementation decreases the proportions of unsaturated fatty acids [150,151].

Moreover, copper deficiency causes hypercholesterolemia. Dietary copper restriction results in low plasma copper levels and low CP and SOD activities but increased plasma free cholesterol in humans [152] and rodents [153,154]. Conversely, supplementation of hypercholesterolemic patients with 5 mg/day copper for 45 days decreased total plasma cholesterol and slightly increased high-density lipoprotein (HDL) cholesterol [155]. It has not been thoroughly defined how copper controls cholesterol homeostasis, but animal studies provide some insights. Diet-induced copper deficiency increases hepatic glutathione (GSH) levels and subsequently increases the activity of cardiac hydroxymethylglutaryl-coenzyme A (HMG-CoA) reductase, which controls the rate-limiting step of cholesterol biosynthesis [153]. Furthermore, copper deficiency significantly increases the total plasma pool of cholesterol due to enlargement of plasma volume. Mice fed a marginal copper-deficient diet (1.6 mg/kg) first show a larger pool of plasma cholesterol at 3 weeks, followed by an increase in plasma cholesterol concentrations at 5 weeks. However, in rats fed a copper-deficient diet (0.6 mg/kg), both elevations of the plasma pool size and cholesterol concentration are detected at 3 weeks [154].

Changes in the composition of lipoprotein components by copper deficiency are another contributor to IHD. In copper-deficient rats, the plasma levels of triglyceride-containing low- and very low-density lipoproteins (LDLs and VLDLs) increased by 1.6- and 2.7-fold, respectively. In addition, copper deficiency increases the susceptibility of LDLs and VLDLs to oxidation [156]. Increased lipoprotein oxidation during copper deficiency is due to increases in the triglyceride content of these lipoproteins [140] and decreases in SOD1 activity [157]. Although copper deficiency increases the expression of the lipoprotein ApoE [145] and HDL cholesterol concentrations [158,159], it does not alter the lipid composition of HDLs. In addition, the activity, but not mRNA expression, of plasma lecithin: cholesterol acyltransferase (LCAT), which catalyzes esterification of free cholesterol in HDLs, is reportedly reduced in copper-deficient rats [160]. However, it is unclear whether these HDLs in the copper-deficient state improve or worsen cholesterol efflux activity. Nonetheless, copper deficiency increases the ratio of atherogenic VLDL and LDL cholesterol to protective HDL cholesterol [158,159]. A better understanding of the molecular mechanism resulting in dysregulation of lipid metabolism caused by copper deficiency will provide new insights to reduce the incidence of IHD.

During numerous attempts to identify the causes of IHD, copper supplementation and adequate dietary copper have been demonstrated to reduce the risk of IHD. In a study of adult men with moderately high cholesterol, 2 mg/day copper supplementation for 4 weeks increased both erythrocyte SOD1 and lipoprotein oxidation lag time, the latter of which is a risk factor for IHD [161]. In another study of adult women with moderate hypercholesterolemia, 2 mg/day copper supplementation for 8 weeks elevated erythrocyte SOD1 and plasma CP levels. However, no significant changes in plasma cholesterol concentrations by copper supplementation were observed, which may be due to the small number of patients recruited and the different diets used. In addition, copper reduces the mean plasma oxidized LDL level (21 of 35 subjects showed a decrease), which helps to lower IHD risk [162]. In healthy young women, copper supplementation (6 mg/day for three 4-week periods with 3-week washouts between periods increased erythrocyte SOD1 activity and decreased fibrinolytic factor plasminogen activator inhibitor type 1 concentrations, suggesting that copper supplementation reduces IHD risks [163]. Antioxidants, particularly carotenoids, are tightly linked to IHD [164]. In humans, low levels of carotenoids are associated with a higher risk of myocardial infarction and the development of atherosclerosis and hypertension, as well as higher levels of circulating inflammatory cytokines [164]. Low circulating carotenoid levels are also associated with high oxidative stress evidenced by decreased circulating SOD levels [165]. Interestingly, in healthy adults, while copper supplementation increased red blood cell hemolysis time, which was positively and significantly correlated with circulating carotenoid levels, the administration of copper chelators reduced levels of circulating carotenoids, including lycopene and carotenes [166]. However, the molecular mechanisms by which copper deficiency promotes IHD and copper supplementation benefits IHD have not been extensively investigated and must be further elucidated.

### 4.3. DM cardiomyopathy

Copper-deficient diet-induced cardiomyopathy is characterized by global decreases in circulating and cardiac copper concentrations. Cu^+^ comprises 95% of total copper and localizes intracellularly, whereas Cu^2+^ comprises 5% of total copper and localizes extracellularly [167]. In contrast to copper deficiency-induced cardiomyopathy, increases in circulating copper concentrations and 2–3-fold increases in extracellular myocardial Cu^2+^ levels, but decreases in intracellular myocardial Cu^+^ levels, were reported in humans and rodents with DM cardiomyopathy. The reduced myocardial copper content and elevated systemic and total cardiac copper content in DM cardiomyopathy reflect defective uptake of copper by myocardiocytes [168,169,170,171]. Glycosylation of proteins to form advanced glycation end-products (AGEs) is a deleterious consequence of hyperglycemia in diabetes and metabolic syndrome [172]. In hearts of rats with DM, increased extracellular Cu^2+^ increases gene expression of *Tgfβ*, *Smad4*, and collagens, which results in collagen deposition and increases the formation of AGEs of collagens. These events cause vascular injury and increase susceptibility to IHD [173]. Elevated extracellular Cu^2+^ in DM cardiomyopathy is likely loosely bound to extracellular matrix components, such as collagens [168,169]. Interestingly, in rodents and humans with DM cardiomyopathy, Cu^2+^-selective chelators, including trientine and triethylenetetramine (TETA) dihydrochloride, prevent excessive cardiac collagen deposition, improve cardiac structure and function, and restore antioxidant defense by promoting copper excretion [168,174,175,176,177,178]. Zhang et al. reported that the expression of the *Ctr1* gene was downregulated in hearts of rats with DM, which is consistent with impaired cardiac copper uptake in DM [176]. Although TETA decreases the expression of cardiac *Ctr1* in rats, it increases CTR2 localization to the plasma membrane and, thus, concomitantly normalizes the reduced cardiac Cu^+^ levels in DM. In addition, TETA increases localization of ATP7A to the TGN and peri-nuclear region and corrects the defects in copper delivery to the secretory pathway and, thus, improves the utilization of copper by cuproenzymes, including ATOX1 and SOD1 [176]. These data suggest that TETA normalizes cardiac copper homeostasis and restores cardiac function in DM by restoring expression and localizations of copper transporters and copper-binding proteins. In addition, TETA restores mRNA and protein expression of copper chaperones, including COX11, COX17, CCS, and SOD1 and, thus, restores copper availability and trafficking, and improves cardiac functions in hearts of rats with DM [175]. Although the highly selective Cu^2+^ chelator trientine efficiently treats DM cardiomyopathy, long-term clinical studies are necessary to determine whether the improvement of cardiac function by trientine is associated with long-term benefits for mortality. Similarly, additional studies investigating the effects of trientine for treatment of other cardiomyopathies, such as IHD, are also warranted. The impacts of copper deficiency on heart disease and results of clinical and preclinical (in vitro and in vivo) studies using copper supplementation and copper chelators to treat heart disease are summarized in Table 2 and Table 3, respectively.

## 5. Conclusions

Heart disease results in immense health and economic burdens worldwide, and its cause and underlying molecular mechanism must be urgently elucidated. Although copper deficiency affects many tissues, including the liver, intestine, vasculature, brain, and adipose tissue, the heart is one of the most sensitive tissues to copper deficiency. Copper plays an important role in maintaining the activities of cardiac copper-dependent proteins, including CCO, SOD1/3, MT, CP, and LOX, which are essential for regulation of oxidative phosphorylation, iron mobilization, antioxidant defense, and connective tissue crosslinking in the heart. Numerous preclinical and clinical studies demonstrate the critical role of copper homeostasis in maintaining cardiac function. Defective copper metabolism, including copper deficiency or defects in copper chaperones, copper transporters, or copper-binding proteins, results in cardiac dysfunction and myopathy and increases the risk of mortality. Copper supplements and copper chelators are used to mitigate the deleterious impact of copper deficiency and to normalize excess extra-myocellular copper in order to treat heart disease. However, much more efforts are needed to determine the efficacy and safety of copper supplementation and copper chelators and their underlying mechanisms.

## Figures and Tables

**Figure 1 nutrients-14-00700-f001:**
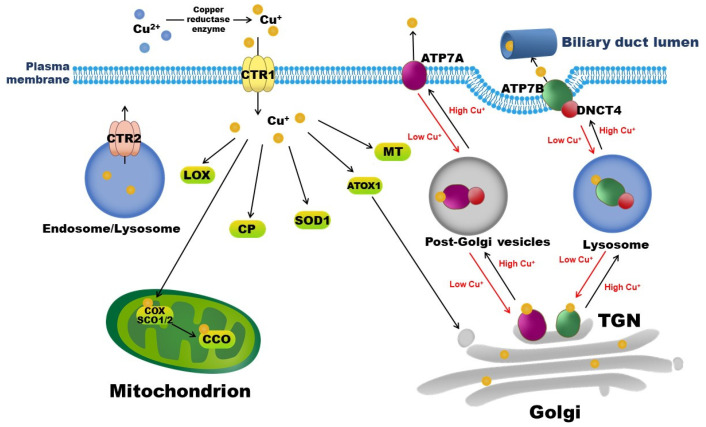
Copper-binding proteins and intracellular copper transportation. Copper is exclusively absorbed by enterocytes in the small intestine via CTR1. CTR2 is a low-affinity copper importer that localizes to endosomes and lysosomes. Intracellular copper-binding proteins include COX, SCO, SOD, MT, CP, and LOX. ATP7A and ATP7B are copper exporters. Under normal conditions, ATP7A and ATP7B localize to TGN, where they supply copper to copper-dependent enzymes in the secretory pathway. When the cytosolic copper level rises, ATP7A or ATP7B interacts with DNCT4 and traffics to endosome-like vesicles and then to the plasma membrane, pumping excess copper into the extracellular space, or into bile in the case of the liver, to reduce intracellular copper levels. By contrast, when the intracellular copper level is low, ATP7A or ATP7B recycles to the TGN and transports copper from the cytoplasm into the Golgi. ATOX1: antioxidant 1 copper chaperone; ATP7A: copper-transporting ATPase 1; ATP7B: copper-transporting ATPase 2; CTR1: copper transporter 1; CTR2: copper transporter 2; CCO: cytochrome c oxidase; COX: cytochrome c oxidase copper chaperone; CP: ceruloplasmin, DNCT4: p62 subunit of dynactin; LOX: lysyl oxidase; MT: metallothionein; SCO: synthesis of cytochrome c oxidase; SOD: superoxide dismutase; TGN: trans-Golgi network.

**Table 1 nutrients-14-00700-t001:** The major findings of copper chaperones/transporters and their regulator.

Study Object	Gene Modification/Mutation	Finding	Reference
Human	SCO2 (E140K)	Fetal/infantile cardiac hypertrophy	[10,37,38,39,40]
Human	SCO1 (G132S)	Early onset cardiac hypertrophy	[41]
Mice	Myocardiocyte-specific *Sco1* knockout	Dilated cardiomyopathy	[42]
Human	COA6 (W66R)	Cardiac hypertrophy	[8]
Human	COA6 (W59C/E87X)	Cardiac hypertrophy	[45]
Mice	*Sod1* knockout	Cardiac injury (apoptosis and inflammation)	[46]
Mice	MT1/2 knockout	Cardiac dysfunction and fibrosis	[47]
Mice	*Sod3* knockout	Cardiac injury (hypertrophy, fibrosis, apoptosis, and inflammation)	[48,49]
Human	SOD3 (R231G)	Positively associated with IHD, myocardial infarction, and HF in diabetic subjects	[50,51,52,53,54]
Human	Rs1307255 variant	Moderately increased circulating CP levels and high circulating CP levels are associated with major adverse cardiovascular events	[55,56]
Mice	Myocardiocyte-specific *Ctr1* knockout	Cardiomyopathy with cardiac hypertrophy and endocardial fibrosis	[5]
Mice	Intestinal-specific *Ctr1* knockout	Cardiac hypertrophy	[57]
Human	ATP7A mutation (Menkes disease)	High frequency of congenital heart disease	[58]

ATP7A: copper-transporting ATPase 1; CP: ceruloplasmin; COA6: cytochrome c oxidase assembly factor 6; *Ctr1*: copper transporter 1; HF: heart failure; IHD: ischemic heart disease; MT: 
metallothionein; SCO: synthesis of cytochrome c oxidase; SOD: superoxide dismutase.

**Table 2 nutrients-14-00700-t002:** Clinical and preclinical copper supplementation and copper chelator treatments.

Study Object	Mutation/Model	Treatment	Treatment Length	Results	Reference
A patient of 25 months old	Homozygous E140K mutation in SCO2	Oral copper supplementation (140 μg/day)	14 months	Improved cardiac hypertrophy and function	[10]
A patient’s fibroblasts	Homozygous W66R mutation in COA6	Copper chloride (25–200 μmol/L)	7 days	Partially restored protein expression levels of subunits of mitochondrial complex IV	[8]
Mice	Ascending aortic constriction	Copper dietary treatment	6 mg/Cu/kg diet for 4 weeks and 20 mg/Cu/kg diet for another 4 weeks	Restored VEGF expression and angiogenesis	[107]
H9C2 cells	Hydrogen peroxide treatment	Copper sulfate (5 μM)	48 h	Suppressed cardiomyocyte hypertrophy	[108]
Mice	Copper-deficient diet from day 3 postdelivery for 4 to 5 weeks	Copper-adequate diet (6 mg/kg) feeding	4 weeks	Restored cardiac diastolic and systolic function	[122]
Hypercholesterolemic patients	Hyperlipidemia	Oral copper supplementation (5 mg/day)	45 days	Decreased total plasma cholesterol and increased HDL cholesterol	[155]
Adult men	Moderate hypercholesterolemia	Oral copper supplementation (2 mg/day)	4 weeks	Increased both erythrocyte SOD1 and lipoprotein oxidation lag time	[161]
Adult women	Moderate hypercholesterolemia	Oral copper supplementation (2 mg/day)	8 weeks	Elevated erythrocyte SOD1 and plasma CP levels	[162]
Yong women	Healthy volunteer	Oral copper supplementation (6 mg/day)	4 weeks	Increased erythrocyte SOD1 activity and decrease fibrinolytic factor plasminogen activator inhibitor type 1 concentrations	[163]
Rats	Type 2 diabetes	Trientine (8–11 mg/day); TETA (20–30 mg/day per rat)	7–8 weeks	Prevented excessive cardiac collagen deposition, improved cardiac structure and function, and restored antioxidant defense	[165,171,172,173]
Diabetic patients	Type 2 diabetes with left ventricular hypertrophy	Trientine (600 mg/day)	12 months	Decreased left ventricular mass indexed to body surface area (LVMbsa)	[174,175]

COA6: cytochrome c oxidase assembly factor 6; CP: ceruloplasmin; HDL: high-density lipoprotein; SCO: synthesis of cytochrome c oxidase; SOD: superoxide dismutase, TETA: triethylenetetramine; VEGF: vascular endothelial growth factor.

**Table 3 nutrients-14-00700-t003:** Copper deficiency and heart diseases.

Disease	Mechanism	Reference
Cardiac hypertrophy	Decreased CCO activity and ATP synthase function and compensatory enlargement of mitochondria and mitochondrial biogenesis	[19,35,100]
HF	Diastolic dysfunction and a blunted response to β-adrenergic stimulation	[120,121,122]
HF	Perturbation of cellular calcium homeostasis	[124,125]
HF	Elevated NO production	[129,130,131]
IHD	Accumulation of free fatty acids	[142,143,144,145,146]
IHD	Changes in fatty acid composition	[148,149]
IHD	Hypercholesterolemia	[152,153,154]
IHD	Alterations in plasma lipoprotein levels and compositions	[140,145,156,157,158,159,160]
DM cardiomyopathy	Increased collagen deposition and the formation of AGEs of collagen	[165,166,170]

AGEs: advanced glycation end-products; ATP: adenosine triphosphate; CCO: cytochrome c oxidase; DM: diabetes mellitus; HF: heart failure; IHD: ischemic heart disease; NO: nitric oxide.

## Data Availability

Not applicable.

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
