# Peer review of "An Emerging Role of Defective Copper Metabolism in Heart Disease"

_nutrients, 2022, doi:10.3390/nu14030700_

Round 1

Reviewer 1 Report

Excellent and very extensive review on the subject.

Author Response

We thank the reviewers for their insightful comments and suggestions. We are pleased that the reviewers found the work to be a significant contribution to the field. We have worked hard to address each of the concerns raised, with one figure, three tables, and three references included in the revised manuscript. The manuscript has been revised to increase the readability by adding one graphic representation of functions of copper chaperones and copper transporters discussed in the review. The revised manuscript also includes three new tables: 1) gathering major findings of the literature on copper-binding protein and heart disease, and 2) listing preclinical and clinical studies on the effects of altering copper content on heart diseases. Moreover, the revised manuscript contains a discussion on the relation of copper homeostasis, antioxidants, and heart disease. All changes are highlighted in blue throughout the manuscript. We hope you agree that the manuscript is now acceptable for publication in Nutrients

Reviewer 2 Report

To:

Editorial Board

Nutrients

Title: “An emerging role of defective copper metabolism in heart disease”

Dear Editor,

I read this paper and I think that:

  • the authors should include representative figures for this paper. This would increase the readability of the text and increased its comprehension by the readers.
  • Furthermore, the authors should include a table gathering the main findings from literature. Please provide.
  • The authors should provide data from literature about possible strategies to counteract alterations in copper metabolism. Data on therapeutic management are fundamental for improving the quality of the paper.
  • For instance, the authors should discuss the role of antioxidants in such a context. They can consider the paper from Ciccone MM et al. Mediators Inflamm. 2013;2013:782137.

Round 2

Reviewer 2 Report

the authors well addressed my previous comments. The paper improved very much